# Derivation of a Viscous Serre–Green–Naghdi Equation: An Impasse?

**Denys Dutykh** [1,†,‡] **and Hervé V.J. Le Meur** [2,*,†]

1 Univ. Grenoble Alpes, Univ. Savoie Mont Blanc, CNRS, LAMA, 73000 Chambéry, France; Denys.Dutykh@univ-smb.fr
2 CNRS, UMR 7352, LAMFA Université Picardie Jules Verne, 80039 Amiens, France
\* Correspondence: Herve.Le.Meur@u-picardie.fr
† These authors contributed equally to this work.
‡ D.D. dedicates this work to his friend and collaborator Prof. Jean-Guy Caputo on the occasion of his 60th birthday.

**Abstract:** In this article, we present the current status of the derivation of a viscous Serre–Green–Naghdi system. For this goal, the flow domain is separated into two regions. The upper region is governed by inviscid Euler equations, while the bottom region (the so-called boundary layer) is described by Navier–Stokes equations. We consider a particular regime binding the Reynolds number and the shallowness parameter. The computations presented in this article are performed in the fully nonlinear regime. The boundary layer flow reduces to a Prandtl-like equation that we claim to be irreducible. Further approximations are necessary to obtain a tractable model.

**Keywords:** Serre–Green–Naghdi; viscous fluid; asymptotic model; nonlinear regime

**PACS:** 47.35.Bb; 47.10.ad; 47.85.Dh; 47.10.A-

**MSC:** 76B15; 76D05; 76D33; 76D10





## 1. Introduction

The water wave theory has been essentially developed in the framework of the inviscid and, very often, irrotational Euler equations. However, various viscous effects are inevitably present in laboratory experiments and even more in the real world. Thus, the conservative conventional models have to be supplemented with dissipative effects to improve the quality of their predictions. A straightforward energy balance asymptotic analysis shows that the main dissipation occurs at the bottom boundary layer [1] [Section §2] (or at the lateral walls if they are also present [2,3]). In this way, the corresponding long wave and small amplitude Boussinesq-type systems have been derived taking into account the boundary layer effects [4]. In [5], the author derived the viscous Boussinesq model without the irrotationality assumption that is usually done for Euler equations. Other articles already took the vorticity into account, even for fully nonlinear Boussinesq equations (here, called Serre–Green–Naghdi or SGN) [6,7] but not yet viscosity.

Physics gives us rather general equations, such as Euler or Navier–Stokes for fluid dynamics. Despite there being only model equations, physicists do believe they are very close to reality, at least for a wide range of fluids and types of experiments (geometry and boundary conditions). One always may solve these general equations for specific conditions. However, if the geometry and scales permit it, solving a reduced model in $1 + 1$D (one dimension in space and one in time) or even $2 + 1$D is better than the full equation $(3 + 1$D$)$. One of the main motivations is to apply such simulations to coastal flow where the coast is clearly 2D. Therefore, taking off one dimension is very valuable. Great efforts are devoted to finding reduced models from these general equations for at least two

reasons. The first one is numerical: if a model is proven to be right up to a certain time and is simpler to simulate, then it may save computation time. Additionally, if the goal is to predict the time at which a tsunami reaches the coasts, hours and even minutes are relevant. The second one is that some reduced model trigger behaviors that cannot be easily predicted from the full model. One of the best examples is the solitary wave phenomenon, which was observed and reported by J.S. Russell (ca. 1845). It was explained independently by J. Boussinesq (1877) and Korteweg–de Vries (1895) using the reduced Korteweg–de Vries (KdV) model, as we call it nowadays. However, the first proof of the existence of solitary waves in the full Euler equations is due to Lavrentyev (ca. 1945). Even later, this solitonic behavior was rediscovered in a seminal work by Zabusky and Kruskal (1965), which opened the whole research direction in infinite-dimensional integrable systems. It is precisely a numerical simulation of this reduced equation that suggests that there exist solutions that do not vanish and even remain invariant (solitary waves). However, to reduce the numerical size of a system, one must find a small parameter and then find the expansion of the system. Therefore, the asymptotic method is a major tool to reduce the models.

Up until now, the inviscid free boundary flow with small fields (almost linear) in only one direction is modeled by a KdV equation and is rather well-known. It uses one field in 1 + 1D instead of eight fields (two velocity components, one pressure, and one surface wave height) in 3 + 1D. When you take viscosity into account, the same equation holds with a half derivative more ([5]). The propagation of a surface wave in any direction, for an inviscid fluid, is proven to be modeled by the Boussinesq system [8] and requires two scalar fields in 1 + 1D. Its viscous counterpart is also a system of 1 + 1D but with a half-derivative and initial conditions in the whole boundary layer. Thus, indeed, it is only a 2 + 1D system. In the inviscid case of a non-small surface wave, an SGN system was derived [9,10]. It was proven to extend the Boussinesq system [8] and to be well-posed [11] and is numerically more stable than the Boussinesq systems [7]. Here, we try to extend the SGN system to take viscosity into account.

In the present article, we report the current status of the attempt to derive a viscous counterpart of the well-known inviscid SGN equations. We chose an asymptotic regime that binds the Reynolds number and the shallowness parameter from the special case of the linear regime. We write the equations, then solve them in the bulk part and then in the boundary layer, and try to match the two. A full derivation appears very unlikely since it would require being able to reduce the Prandtl's equation found in the boundary layer.

## 2. Primary Equations

Consider the flow of an incompressible viscous liquid in a physical two-dimensional space over a flat bottom and with a free surface. We assume additionally that the fluid is homogeneous (i.e., density $\rho$ is constant) and that the gravity acceleration $g$ is constant. For the sake of simplicity, in this study, we neglect all other forces (such as the Coriolis force and friction). Hence, we deal with pure viscous gravity waves. We introduce a Cartesian coordinate system $O\,\tilde{x}\,\tilde{y}$. The horizontal line $O\,\tilde{x}$ coincides with the still water level $\tilde{y} = 0$, and the axis $O\,\tilde{y}$ points vertically upwards. The fluid layer is bounded below by the horizontal solid bottom $\tilde{y} = -d$ and above by the free surface $\tilde{y} = \tilde{\eta}\,(\tilde{x},\,\tilde{t})$. The sketch of the fluid domain is shown in Figure 1.

In order to make the equations dimensionless, we choose a characteristic horizontal length $\ell$ (characteristic wavelength of the surface wave), vertical height of the free surface $A$ (departure from the mean position of the free boundary), and mean depth $d$. All this enables us to define a characteristic velocity $c_0 = \sqrt{gd}$. Then, one may define the dimensionless independent variables:

$$\tilde{x} = \ell x,\ \tilde{y} = dy,\ \tilde{t} = t\ell/c_0.$$

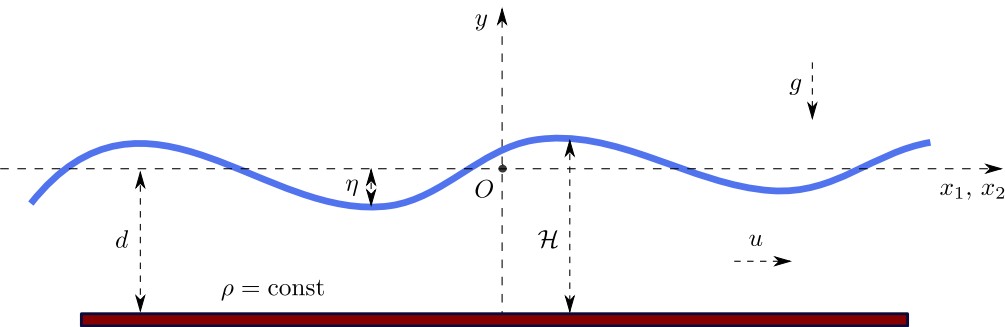

**Figure 1.** Sketch of the fluid domain.

This enables us to define the dimensionless fields:

$$\tilde{u} = c_0 u, \ \tilde{v} = \frac{d c_0}{\ell} v, \ \tilde{p} = \tilde{p}_{\text{atm}} - \rho g d\, y + \rho g d\, p, \ \tilde{\eta}(\tilde{x}, \tilde{y}, \tilde{t}) = A\eta(x, y, t).$$

We also define some dimensionless numbers, characteristic of the flow:

$$\varepsilon = \frac{A}{d}, \ \mu^2 = \frac{d^2}{\ell^2}, \ \text{Re} = \frac{\rho c_0 l}{\nu}.$$

The system of Navier–Stokes equations can then be written in 2D and in dimensionless variables:

$$\begin{cases}
u_t + u u_x + v\, u_y - \left( u_{xx} + u_{yy}/\mu^2 \right)/\text{Re} + p_x = 0 \\[4pt]
\mu^2 (v_t + u v_x + v v_y) - \left( \mu^2 v_{xx} + v_{yy} \right)/\text{Re} + p_y = 0 \\[4pt]
u_x + v_y = 0 \\[4pt]
\left[ -(p - \varepsilon\eta)\boldsymbol{I} + \dfrac{2}{\mu\text{Re}} \begin{pmatrix} \mu u_x & (u_y + \mu^2 v_x)/2 \\ (u_y + \mu^2 v_x)/2 & \mu v_y \end{pmatrix} \right]\Bigg|_{\varepsilon\eta} \boldsymbol{n} = 0 \quad \text{on } y = \varepsilon\eta \\[4pt]
\eta_t + u(y = \varepsilon\eta)\eta_x - v(y = \varepsilon\eta)/\varepsilon = 0 \quad \text{on } y = \varepsilon\eta \\[4pt]
u(y = -1) = v(y = -1) = 0,
\end{cases} \tag{1}$$

where we denote $u|_{\varepsilon\eta} = u(y = \varepsilon\eta) = u(x, y = \varepsilon\eta(x, t), t)$ and the normal $\boldsymbol{n} = (-\varepsilon\eta_x, 1)/\sqrt{1 + \varepsilon^2 \eta_x^2}$.

One could assume the fields to be small around the hydrostatic flow (which is lifted by the change in field from $\tilde{p}$ to $p$), so around $(u, v, p, \eta) \simeq 0$. However, such an assumption would be too particular for our nonlinearity assumption, which reads $\varepsilon = O(1)$. Since the linear case $\varepsilon = o(1)$ must be satisfied also, we are justified in making this assumption. Then, we would be led to a linear system identical (up to changes of variables) to System (7) of [5]. The study of this linear system is rather arduous and without interest to reproduce here. In the end, even though linearity is not the regime assumed here, it must be included in our study as a special case (if $\varepsilon \to 0$). It suggests then to assume:

$$\text{Re} \sim \mu^{-6}. \tag{2}$$

Below, we solve the problem in the bulk part where Euler's equations are justified to apply (Section 2.1) and, then, we try to solve the velocity in the boundary layer (Section 2.2). In this last section, we are led to Prandtl's equation that prohibits any further advance to the best of our knowledge. The sketch of the boundary layer fluid domain is depicted in Figure 2.

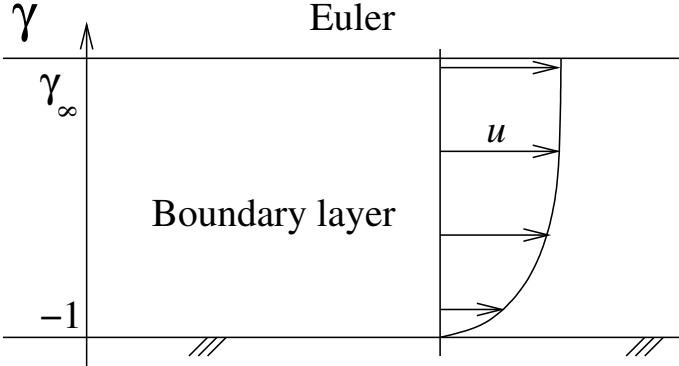

**Figure 2.** The sketch of the boundary layer domain.

What is the size of the boundary layer where the no-slip condition forces the fluid to have a large gradient of velocity? In the same way as in [5], one may assume it is of size $\mu^2$. Inside the boundary layer, we make a change in the vertical variable (justified below):

$$y = -1 + \mu^2 \gamma. \tag{3}$$

Then, we assume $\gamma$ to be nonnegative and up to "large" values. One may use two strategies. The first sets an upper bound. Then, one writes the matching condition at that given height. However, one might wonder whether the artificial choice of this upper bound does restrict the result. The second method sets an artificial upper bound $\gamma_\infty$. Then, the matching condition is written at that height and one must check that the resulting condition does not depend on this $\gamma_\infty$. Below, we use the latter method and thus $\gamma \in [0, \gamma_\infty)$. Our $\gamma_\infty$ is large but not so large so as to let $\mu^2 \gamma_\infty \ll 1$. However, let us start with the bulk movement in the upper part.

### 2.1. Resolution in the Upper Part (Euler)

In the upper part, $y \gg -1 + \mu^2$ and $\mu^4$ are small. Therefore, one may drop the Laplacian and keep the following from (1):

$$
\begin{cases}
u_t + uu_x + vu_y + p_x = O\left(\dfrac{u_{yy}}{\mu^2 \text{Re}}\right) + O(\mu^6) & \\
\mu^2(v_t + uv_x + vv_y) + p_y = O(\mu^6) & \\
u_x + v_y = 0 & \\
-p + \varepsilon\eta = O\left(\dfrac{u_y|_{\varepsilon\eta}}{\mu \text{Re}}\right) + O\left(\dfrac{1}{\text{Re}}\right) & \text{on } y = \varepsilon\eta \\
(p - \varepsilon\eta)\varepsilon\eta_x + 2(-u_x\varepsilon\eta_x + (u_y/\mu + \mu v_x))/\text{Re} = 0 & \text{on } y = \varepsilon\eta \\
\varepsilon(\eta_t + u|_{\varepsilon\eta}\eta_x) = v|_{\varepsilon\eta} & \text{on } y = \varepsilon\eta.
\end{cases}
\tag{4}
$$

Here and below, for $f = u$ or $v$, we denote $f|_{\varepsilon\eta} = f(x, y = \varepsilon\eta(x,t), t)$. First, one may notice that the viscosity terms are no more present inside this part of the domain. It is argued in [5] that one may (and even must) drop the fifth equation from this system due to the fact that the fluid is indeed no more viscous in this part of the domain.

It is classical to use (4)$_3$ to get

$$v = v|_{\varepsilon\eta} - \int_{\varepsilon\eta}^{y} u_x \, \mathrm{d}y', \tag{5}$$

where $v|_{y=\varepsilon\eta}$ is given by (4)$_6$. One may use this vertical velocity in (4)$_2$ to compute $p_y$. Thanks to (4)$_4$, one has

$$p = \varepsilon\eta - \mu^2\left[(y-\varepsilon\eta)\left((v|_{\varepsilon\eta})_t + u_x|_{\varepsilon\eta}\varepsilon\eta_t\right) + \left(\int_{\varepsilon\eta}^y u\right)\left((v|_{\varepsilon\eta})_x + u_x|_{\varepsilon\eta}\varepsilon\eta_x\right)\right.$$
$$-\left(\int_{\varepsilon\eta}^y u_x\right)v|_{\varepsilon\eta} - \int_{\varepsilon\eta}^y\int_{\varepsilon\eta}^{y'} u_{xt} - \int_{\varepsilon\eta}^y\left(u\int_{\varepsilon\eta}^{y'} u_{xx}\right) + \left.\int_{\varepsilon\eta}^y\left(u_x\int_{\varepsilon\eta}^{y'} u_x\right)\right]$$
$$+ O\left(\frac{u_y|_{\varepsilon\eta}}{\mu\mathrm{Re}}\right) + O\left(\frac{1}{\mathrm{Re}}\right) + O(\mu^6). \tag{6}$$

Thus, we have both $v$ (thanks to (5)) and $p$ (thanks to (6)) and may rewrite (4)$_1$ with the only fields $u$ and $\eta$:

$$u_t + uu_x + u_y\left(v|_{\varepsilon\eta} - \int_{\varepsilon\eta}^y u_x\right) + \varepsilon\eta_x - \mu^2\left[(y-\varepsilon\eta)\left((v|_{\varepsilon\eta})_t + u_x|_{\varepsilon\eta}\varepsilon\eta_t\right)\right.$$
$$+ \left(\int_{\varepsilon\eta}^y u\right)\left((v|_{\varepsilon\eta})_x + u_x|_{\varepsilon\eta}\varepsilon\eta_x\right) - \left(\int_{\varepsilon\eta}^y u_x\right)v|_{\varepsilon\eta} - \int_{\varepsilon\eta}^y\int_{\varepsilon\eta}^{y'} u_{xt}$$
$$\left.- \int_{\varepsilon\eta}^y\left(u\int_{\varepsilon\eta}^{y'} u_{xx}\right) + \int_{\varepsilon\eta}^y\left(u_x\int_{\varepsilon\eta}^{y'} u_x\right)\right]_x = O\left(\frac{(u_y|_{\varepsilon\eta})_x}{\mu\mathrm{Re}}\right) + O\left(\mu^6\right) + O\left(\frac{u_{yy}}{\mu^2\mathrm{Re}}\right). \tag{7}$$

To take off the dependence on $y$ of this equation, we integrate between the bottom of our upper part ($y = -1 + \mu^2\gamma_\infty$) and its upper free boundary ($y = \varepsilon\eta(x,t)$). We define the following:

$$\mathcal{H}_{\mu,\gamma_\infty} = 1 + \varepsilon\eta - \mu^2\gamma_\infty, \text{ and } \bar{u}(x,t) = \frac{1}{\mathcal{H}_{\mu,\gamma_\infty}}\int_{-1+\mu^2\gamma_\infty}^{\varepsilon\eta(x,t)} u(x,y)\,\mathrm{d}y. \tag{8}$$

We also need a lemma that will enable us to commute the integration and the $x$ differentiation under an assumption:

**Lemma 1.** *Let $F$ be a $C^1$ function defined in $\Omega = \{(x,y)/x \in \mathbb{R}, -1 + \mu^2\gamma_\infty < y < \varepsilon\eta(x)\}$, such that, if $\forall x$, $F(x, y = \varepsilon\eta) = 0$, then*

$$\int_{-1+\mu^2\gamma_\infty}^{\varepsilon\eta}\frac{\partial F}{\partial x}(x,y)\mathrm{d}y = \frac{\partial}{\partial x}\int_{-1+\mu^2\gamma_\infty}^{\varepsilon\eta} F(x,y)\mathrm{d}y. \tag{9}$$

The proof is very simple and left to the interested reader. We apply it to (7) because the $x$ differentiation of functions inside the square brackets commutes with our integral. Indeed, every function inside the brackets vanishes at $y = \varepsilon\eta$.

Thanks to Lemma 1, one may commute the $x$ differentiation of the square bracket in Equation (7) with the integral since the terms in the square brackets vanish at $y = \varepsilon\eta$. An integration by parts of the integral $\int u_y\left(v|_{\varepsilon\eta} - \int_{\varepsilon\eta}^y u_x\right)\mathrm{d}y$ term, and the treatment of $\int (u^2)_x$ leads to the following (below, we write $\mathcal{H} = \mathcal{H}_{\mu,\gamma_\infty}$):

$$\mathcal{H}\bar{u}_t + \left(\int_{-1+\mu^2\gamma_\infty}^{\varepsilon\eta} u^2\right)_x + \mathcal{H}\mathcal{H}_x + (\bar{u} - u|_{-1+\mu^2\gamma_\infty})(\mathcal{H}_t + (\mathcal{H}\bar{u})_x) - \bar{u}(\mathcal{H}\bar{u})_x$$
$$- \mu^2\left[-\frac{\mathcal{H}^2}{2}\left((\partial_t + \bar{u}\partial_x)(v|_{\varepsilon\eta}) + \mathcal{H}_t(u_x|_{\varepsilon\eta} - \bar{u}_x) + \mathcal{H}_x(\bar{u}u_x|_{\varepsilon\eta} - \bar{u}_x u|_{\varepsilon\eta})\right)\right.$$
$$+ \int_{-1+\mu^2\gamma_\infty}^{\varepsilon\eta}\int_{\varepsilon\eta}^y (u-\bar{u})\mathrm{d}y'\,\mathrm{d}y \times \left((v|_{\varepsilon\eta})_x + u_x|_{\varepsilon\eta}\mathcal{H}_x\right) - \int_{-1+\mu^2\gamma_\infty}^{\varepsilon\eta}\int_{\varepsilon\eta}^y (u-\bar{u})_x\mathrm{d}y'\,\mathrm{d}y\,v|_{\varepsilon\eta}$$
$$\left.- \int_{-1+\mu^2\gamma_\infty}^{\varepsilon\eta}\int_{\varepsilon\eta}^y\left[\int_{\varepsilon\eta}^{y'} u_{xt} + u\int_{\varepsilon\eta}^{y'} u_{xx} - u_x\int_{\varepsilon\eta}^{y'} u_x\right]\mathrm{d}y'\mathrm{d}y\right]_x$$
$$= O\left(\frac{(u_y|_{\varepsilon\eta})_x}{\mu\mathrm{Re}}\right) + O\left(\frac{u_{yy}}{\mu^2\mathrm{Re}}\right). \tag{10}$$

We need now the following (two-fold) assumption for $y \in (-1 + \mu^2 \gamma_\infty, \varepsilon \eta]$:

$$u(x, y, t) = \bar{u}(x, t) + \mu^2 \tilde{u}(x, y, t), \text{ and } \int_{-1+\mu^2\gamma_\infty}^{\varepsilon\eta} \tilde{u} \mathrm{d}y = 0. \tag{11}$$

where $\mathcal{H} = \mathcal{H}_{\mu, \gamma_\infty}$ and $\bar{u}$ is already defined in (8). Notice that the expansion of a function around its mean value $\bar{u}$ is not an assumption. A first way to see this assumption is that the discrepancy with the mean $\bar{u}$ is $\mu^2 \tilde{u}$ and is small ($O(\mu^2)$). Another way to formulate this assumption is to look at an expansion in $\mu^2$, in which one assumes that the zeroth-order term does not depend on $y$ and that the next order term is a zero-mean value. Therefore, $\tilde{u}$ is perpendicular to $\bar{u}$. Whatever the interpretation, the consequence of this assumption is that $O(\mu^2)$ (cross) terms in $\int u^2$ vanish. This gives two different ways to see the real assumptions behind (8) and (11). Finally, this assumption is proven to be true in Lemma 11 (Equation (77)) of [5] in the case of a Boussinesq flow (where $\varepsilon$ is small) *without the assumption of irrotationality* in the Euler part of the flow. The horizontal velocity's expansion is computed in the inviscid case:

$$u(x, y, t) = \int_0^1 u + \mu^2 \eta_{xt}(y^2 - 1/3)/2 + O(\mu^4). \tag{12}$$

Thus, the function is indeed the sum of its mean and an order 2 (in $\mu$) function of mean vanishing. We remind the reader that we still assume that we solve the Euler equations and not yet the Navier–Stokes ones in this part. Therefore, the assumption is coherent with the present derivation.

Upon this assumption, (10) simplifies to

$$\mathcal{H}\bar{u}_t + \mathcal{H}\bar{u}\bar{u}_x + \mathcal{H}\mathcal{H}_x + (\bar{u} - u|_{-1+\mu^2\gamma_\infty})(\mathcal{H}_t + (\mathcal{H}\bar{u})_x)$$
$$- \mu^2 \left[ -\frac{\mathcal{H}^2}{2}(\partial_t + \bar{u}\partial_x)(v|_{\varepsilon\eta}) - \frac{\mathcal{H}^3}{6}(\bar{u}_{xt} + \bar{u}\bar{u}_{xx} - (\bar{u}_x)^2) \right]_x$$
$$= O\left(\frac{(u_y|_{\varepsilon\eta})_x}{\mu \mathrm{Re}}\right) + O\left(\frac{u_{yy}}{\mu^2 \mathrm{Re}}\right) + O(\mu^4). \tag{13}$$

One must add (4)$_6$:

$$v|_{\varepsilon\eta} = \varepsilon(\eta_t + u|_{\varepsilon\eta}\eta_x) = \mathcal{H}_t + u|_{\varepsilon\eta}\mathcal{H}_x = (\partial_t + \bar{u}\partial_x)\mathcal{H} + O(\mu^2) \tag{14}$$

and look for an equation for $u|_{-1+\mu^2\gamma_\infty}$.

**Remark 1.** *The attention may be drawn to the fact that, thanks to (11),*

$$\mathcal{H}_t + (\mathcal{H}\bar{u})_x = \varepsilon\eta_t + \mathcal{H}_x\bar{u} + \mathcal{H}\bar{u}_x = v|_{\varepsilon\eta} + \mathcal{H}\bar{u}_x + O(\mu^2) = v|_{-1+\mu^2\gamma_\infty} + O(\mu^2),$$

*where $v|_{\varepsilon\eta}$ is given by (4)$_6$. In the pure Euler case with no boundary layer ($\gamma_\infty = 0$), $v|_{-1+\mu^2\gamma_\infty} = 0$ since the flow does not cross the boundary. Therefore, we would not need to compute $u|_{-1+\mu^2\gamma_\infty}$. We would have derived the classical SGN equation. We need to go further to obtain a closure of $u|_{-1+\mu^2\gamma_\infty}$.*

### 2.2. Resolution in the Boundary Layer

We write the system that applies in the layer, extracted from (1):

$$\begin{cases} u_t + uu_x + vu_y - u_{xx}/\mathrm{Re} - u_{yy}/(\mu^2\mathrm{Re}) + p_x = 0, \\ \mu^2(v_t + uv_x + vv_y) - \mu^2 v_{xx}/\mathrm{Re} - v_{yy}/\mathrm{Re} + p_y = 0, \\ u_x + v_y = 0, \\ u(y = -1) = v(y = -1) = 0. \end{cases} \tag{15}$$

This system may be rewritten with the change in variables justified in (3) $y = -1 + \mu^2 \gamma$, where $\gamma$ is positive and up to a large (but not too large) $\gamma_\infty$. This change in variable is elicited by the $u_{yy}/(\mu^2 \mathrm{Re}) = u_{yy}\mu^4$ term. The change in variable to an order two in $\mu$ raises this term to order zero. We also use the assumption (2) on Re such that $\mathrm{Re} = R\mu^{-6}$, where $R$ is a constant. We should have tilded the fields but would have dropped the tilde soon after. Therefore, we omit them. When precision is needed, we denote $u^{BL} = u(x, \gamma, t)$ as the horizontal velocity in the boundary layer. The system writes the following:

$$\begin{cases} u_t + u\,u_x + v\,u_\gamma/\mu^2 - u_{xx}R^{-1}\mu^6 - u_{\gamma\gamma}/R + p_x = 0, \\ \mu^2\left(v_t + u\,v_x + v\,v_\gamma/\mu^2\right) - v_{xx}R^{-1}\mu^8 - v_{\gamma\gamma}R^{-1}\mu^2 + p_\gamma/\mu^2 = 0, \\ u_x + v_\gamma/\mu^2 = 0, \\ u(x, \gamma = 0, t) = v(x, \gamma = 0, t) = 0. \end{cases} \tag{16}$$

As is classical, we first compute $v$ (owing to (16)$_3$ and (16)$_4$:

$$v(x, \gamma, t) = 0 - \mu^2 \int_{\gamma=0}^{\gamma} u_x \, d\gamma'. \tag{17}$$

Then, we can compute the differentiated pressure from (16)$_2$ that proves $p_\gamma = O(\mu^4)$. As a consequence,

$$p^{BL}(x, \gamma, t) = p^{BL}(\gamma \to \gamma_\infty) + O(\mu^4),$$

where $p^{BL}(\gamma \to \gamma_\infty)$ is determined thanks to a matching condition at the bottom of the upper part (Euler part). From (6) and owing to the already stated assumption (11), the pressure in the boundary layer is as follows, up to $O(\mu^4)$:

$$p^{Euler}|_{-1+\mu^2\gamma_\infty} = \varepsilon\eta - \mu^2\left[-\mathcal{H}(\partial_t + \bar{u}\partial_x)(v|_{\varepsilon\eta}) - \mathcal{H}^2/2(\bar{u}_{xt} + \bar{u}\bar{u}_{xx} - \bar{u}_x\bar{u}_x)\right] + O(\mu^4). \tag{18}$$

Then, one has the pressure in the boundary layer:

$$p^{BL}(x, \gamma, t) = \varepsilon\eta(x, t) + \mu^2\left[\mathcal{H}(\partial_t + \bar{u}\partial_x)(v|_{\varepsilon\eta}) + \mathcal{H}^2/2(\bar{u}_{xt} + \bar{u}\bar{u}_{xx} - \bar{u}_x\bar{u}_x)\right] + O(\mu^4). \tag{19}$$

Last, we may gather $v^{BL}$ (from (17)) and $p^{BL}$ (from (19)) and rewrite (16)$_1$:

$$u_t^{BL} + u^{BL}u_x^{BL} - u_\gamma^{BL}\int_0^\gamma u_x^{BL}(\gamma')d\gamma' - \frac{u_{\gamma\gamma}^{BL}}{R} + \varepsilon\eta_x$$
$$+ \mu^2\left[\mathcal{H}(\partial_t + \bar{u}\partial_x)^2(\mathcal{H}) + \mathcal{H}^2/2(\bar{u}_{xt} + \bar{u}\bar{u}_{xx} - \bar{u}_x\bar{u}_x)\right]_x = O(\mu^4). \tag{20}$$

At this stage of the derivation, we recognize a Prandtl's equation. It is then intuitive to assume the continuity relation on the horizontal velocity:

$$(u^{BL\infty} =)u^{BL}(x, \gamma_\infty, t) = u^{Euler}(x, z = -1 + \mu^2\gamma_\infty, t)(= u|_{-1+\mu^2\gamma_\infty}).$$

In the Boussinesq regime, the author of [5] had a heat equation (instead of Prandtl's equation) on $u^{BL}$. It was solved with this (upper) boundary condition and the condition at $z = 0$. Once the horizontal velocity (in the boundary) was determined, the author derived the vertical velocity. Then, the continuity condition on the vertical velocity gave the supplementary equation that closed the viscous Boussinesq system.

In our regime, we are led to a system (13) and (20) with boundary conditions depending on $\mathcal{H}, u^{BL}, \bar{u}$. However, the nonlinearity is Prandtl-like and it still depends on $\gamma$ in a hopeless way because of the Prandtl term. Indeed, it is well-known that Prandtl's equation still resists the best physicists and mathematicians. It was proven to be ill-posed in [12] and partially well-posed later. At the current stage, we do not know how to derive a simpler model without unrealistic assumption. At this level, we could not see any more than the two following possible routes:

1. One may use the assumption, classical in the boundary layer community, that the profile is exponential of the type:

$$u^{BL}(x, \gamma, t) = u^{BL\infty}(x, t)(1 - \exp(-\gamma)), \tag{21}$$

which vanishes at $\gamma = 0$ (see (16)$_4$). Such a dependence on $x$ and $\gamma$ is assumed to be split, and this last assumption is very strong. Indeed, (20) may then be rewritten:

$$u_t^{BL\infty}(1 - \exp(-\gamma)) + u^{BL\infty} u_x^{BL\infty}(1 - \exp(-\gamma))^2$$

$$+ u^{BL\infty} \exp(-\gamma) u_x^{BL\infty}(\gamma + \exp(-\gamma) - 1) - \frac{u^{BL\infty}}{R} \exp(-\gamma)$$

$$+ \varepsilon \eta_x + \mu^2 \Big[ \mathcal{H}(\partial_t + \bar{u}\partial_x)(v|_{\varepsilon\eta}) + \mathcal{H}^2/2(\bar{u}_{xt} + \bar{u}\bar{u}_{xx} - \bar{u}_x\bar{u}_x) \Big]_x = O(\mu^4).$$

This equation simplifies to the following:

$$u_t^{BL\infty} + u^{BL\infty} u_x^{BL\infty} + \varepsilon \eta_x + \mu^2 \Big[ \mathcal{H}(\partial_t + \bar{u}\partial_x)(v|_{\varepsilon\eta}) + \mathcal{H}^2/2(\bar{u}_{xt} + \bar{u}\bar{u}_{xx} - \bar{u}_x\bar{u}_x) \Big]_x$$

$$= O(\mu^4) + O(\gamma e^{-\gamma}). \tag{22}$$

Therefore, this first idea gets rid of the second-order derivative that came from the Laplacian. As a consequence, the viscosity is no longer taken into account and it is a deadlock. The error is to assume that the two dependences ($x$ and $\gamma$) are not tied together. Therefore, we may not make such an assumption. The exact shape of the $(x, t)$ dependence in the Boussinesq approximation is given in [5] and recalled below.

2. One may assume the profile of the horizontal velocity in the boundary layer to be the one *proven* in [5] that is a convolution in time mixing $x$ and $\gamma$:

$$u^{BL}(x, \gamma, t) = u^{Euler}(x, z = \varepsilon\gamma_\infty, t) - u^{Euler}(x, \varepsilon\gamma_\infty, .) * \mathcal{L}^{-1}(e^{-\sqrt{R}\sigma\gamma})$$

$$+ \frac{\sqrt{R}}{2} \int_0^{+\infty} (u^{BL,0}(x, \gamma') - u^{Euler,0}(x, z = \varepsilon\gamma_\infty)) \frac{e^{-\frac{R(\gamma'-\gamma)^2}{4t}}}{\sqrt{\pi t}} d\gamma'$$

$$- \frac{\sqrt{R}}{2} \int_0^{+\infty} (u^{BL,0}(x, \gamma') - u^{Euler,0}(x, z = \varepsilon\gamma_\infty)) \frac{e^{\frac{-R(\gamma'+\gamma)^2}{4t}}}{\sqrt{\pi t}} d\gamma' + O(\varepsilon), \tag{23}$$

where $R$ is a constant, $p$ is the dual variable of time $t$ through Laplace transform $\mathcal{L}$, and $\sigma$ is its only root with a nonnegative real part of $p$. However, the Boussinesq assumptions are incompatible with the ones we would do here and such a function would be untractable in Prandtl's equation.

In the system (13) and (20), the $v|_{\varepsilon\eta}$ term must be found from the Euler part. Owing to Euler Equation (4)$_6$, one knows $v|_{\varepsilon\eta} = \mathcal{H}_t + u|_{\varepsilon\eta}\mathcal{H}_x$ (see (14)). Finally, the system (in $\mathcal{H}, \bar{u}, u^{BL}$) to which we are led is no better than

$$\begin{cases} \mathcal{H}\bar{u}_t + \mathcal{H}\bar{u}\bar{u}_x + \mathcal{H}\mathcal{H}_x + (\bar{u} - u^{BL\infty})(\mathcal{H}_t + (\mathcal{H}\bar{u})_x) \\ + \mu^2 \Big[ \frac{\mathcal{H}^2}{2}(\partial_t + \bar{u}\partial_x)^2(\mathcal{H}) + \frac{\mathcal{H}^3}{6}(\bar{u}_{xt} + \bar{u}\bar{u}_{xx} - (\bar{u}_x)^2) \Big]_x = O\left(\frac{(u_y|_{\varepsilon\eta})_x}{\mu\text{Re}}\right) \\ \hspace{3cm} + O\left(u_{yy}/(\mu^2\text{Re})\right) + O(\mu^4) \\ u_t^{BL} + u^{BL} u_x^{BL} - u_\gamma^{BL} \int_0^\gamma u_x^{BL}(\gamma') d\gamma' - \frac{u_{\gamma\gamma}^{BL}}{R} + \varepsilon\eta_x \\ + \mu^2 \Big[ \mathcal{H}(\partial_t + \bar{u}\partial_x)^2(\mathcal{H}) + \mathcal{H}^2/2(\bar{u}_{xt} + \bar{u}\bar{u}_{xx} - \bar{u}_x\bar{u}_x) \Big]_x = O(\mu^4) \\ \hspace{3cm} u^{BL}(\gamma = 0) = 0 \text{ and } u^{BL} \to_{\gamma \to +\infty} u^{BL\infty}. \end{cases} \tag{24}$$

It still depends on $\gamma$ in addition to $x, t$ and requires initial conditions in the whole boundary layer (for all $x$ and $\gamma$). We expected a one-dimensional in space and one dimen-

sional in time two-field ($\bar{u}$ and $\mathcal{H}$) system as is classical for inviscid Boussinesq, viscous Boussinesq [5], and inviscid Serre–Green–Naghdi [9,10,13]. Therefore, we could not reach a reduced enough model.

The first equation of (24) is not surprising. It contains $\bar{u}_t$ and a third-order space derivative of $\mathcal{H}$ as in the inviscid SGN system [9,10]. All the terms are the same as in the classical SGN system, apart from a coupling term depending on $u^{BL\infty}$. $u^{BL\infty}$ is the upper boundary of the horizontal velocity in the boundary layer $u^{BL}$ which must satisfy a Prandtl equation. When one takes viscosity into account from the Boussinesq system, one obtains only two additional terms. One is a half derivative (in time), and the other depends on the initial conditions in the boundary layer. In the nonlinear case we are currently studying, we would have appreciated not having more complex terms. In (24)$_2$, the coupling is performed through $u^{BL\infty}$ and the resolution of a Prandtl equation. Thus, we do need the initial condition in the boundary layer and the resolution of the Prandtl equation in that layer. This is not the goal of the reduced model.

As was said above, in the closest case (the linear viscid case), (24)$_2$ is a parabolic heat-like equation on $u^{BL}$. It may be solved explicitly, and it provides the $u^{BL}(x, \gamma, t)$ given in (23) (see [5]). This $u^{BL}$ enables us to compute the vertical velocity $v^{BL}$ thanks to (17). Then, writing its matching condition at the frontier of the boundary layer with the vertical velocity computed in the Euler (inviscid) part provides a new condition. After some computations, this last condition writes $\eta_t + \bar{u}_x + \bar{u}\,\bar{u}_x \ldots$! Therefore, the nonlinear viscous case lags behind the linear viscous case and the derivation is currently blocked at the Prandtl's step. Would it be possible to further simplify such a coupled Prandtl system similarly to the classical one? This is the main question left by the current study.

## 3. Conclusions

Our goal was to take viscosity and nonlinearity into account for a reduced model of surface gravity waves. As is the case in inviscid fluids, we expected a model depending on the reduced number of dimensions $(1 + 1)$ so that it might be extended easily to the 2D case for numerical simulation purposes. We are stopped in our derivation, in the fully nonlinear regime, at system (24) with initial conditions and $2 + 1$ dimension. We would like to stress that Equations (13) and (20) are still Galilean-invariant despite the presence of the boundary layer. The proposed model enjoys this property because we did not introduce any drastic simplifications yet at this level. To make further progress, the Prandtl-type equation should be further simplified but it seems highly speculative. One strategy could consist in assuming a particular profile of the velocity $u^{BL}$ in the coordinate $\gamma$ similar to the one computed in [5] in the Boussinesq regime, but it is incoherent in our present regime. Further research is needed to reach an effective 1D model if possible.

*Perspectives*

First, one has to resolve the issues mentioned in the present manuscript. The present work opens even more directions. For instance, a referee suggested that the no-slip boundary condition could be relaxed to a slip condition. As the next step, we shall test this relaxation in the Boussinesq regime first, before trying to transpose it to the fully nonlinear models.

**Author Contributions:** Both authors contributed equally to this work. Both authors have read and agreed to the published version of the manuscript.

**Funding:** The work of D.D. has been supported by the French National Research Agency, through Investments for Future Program (ref. ANR-18-EURE-0016—Solar Academy). This research received no other external funding.

**Institutional Review Board Statement:** Not applicable.

**Informed Consent Statement:** Not applicable.

**Data Availability Statement:** Not aplicable.

**Acknowledgments:** First, the authors would like to thank the anonymous referees for their precious help in improving the readability of our manuscript. H.L.M. would like to thank LAMA UMR 5127 for the hospitality during his visits, and D.D. would like to thank LAMFA UMR 7352 for the hospitality during his visits.

**Conflicts of Interest:** The authors declare no conflict of interest.

## Nomenclature

| | |
|---|---|
| $O$ | The origin of the chosen Cartesian coordinate system. |
| $x$ | Horizontal Cartesian coordinate. |
| $\partial_x$ | The partial derivative with respect to $x$. |
| $y$ | Vertical Cartesian coordinate. |
| $\gamma$ | Scaled vertical variable. |
| $\gamma_\infty$ | Upper limit of the boundary layer. |
| $t$ | Time variable. |
| $\partial_t$ | The partial derivative with respect to $t$. |
| $d$ | Undisturbed water depth. |
| $\rho$ | Constant fluid density. |
| $g$ | Gravity acceleration. |
| $\nu$ | The fluid kinematic viscosity. |
| $\boldsymbol{n}$ | The outer unit normal to the free surface. |
| $\ell$ | Characteristic horizontal length. |
| $A$ | Typical wave amplitude. |
| $\eta$ | Free surface elevation above the undisturbed water level. |
| $\mathcal{H}$ | The total water depth. |
| $\Omega$ | The inviscid fluid domain. |
| $c_0$ | Line velocity of infinitely long gravity waves. |
| $u$ | Fluid particle horizontal velocity. |
| $u^{\text{Euler}}$ | The same as above. |
| $\bar{u}$ | The depth-averaged horizontal velocity. |
| $\tilde{u}$ | The deviation of the horizontal velocity from its depth-averaged profile. |
| $u^{BL}$ | The horizontal velocity in the boundary layer. |
| $v^{BL}$ | The vertical velocity in the boundary layer. |
| $u^{BL\infty}$ | The horizontal velocity limit when we approach the boundary layer boundary from the bottom. |
| $v$ | The fluid particle vertical velocity. |
| $p$ | The fluid pressure. |
| $p^{BL}$ | The fluid pressure in the boundary layer. |
| Re | The dimensionless Reynolds number. |
| $\epsilon$ | The dimensionless nonlinearity parameter. |
| $\mu^2$ | The dimensionless dispersion parameter. |

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
