# Peer review of "Derivation of a Viscous Serre–Green–Naghdi Equation: An Impasse?"

_fluids, doi:10.3390/fluids6040135_

Round 1

Reviewer 1 Report

I have reviewed this article before. The authors have made some changes so that the the presentation of the manuscript is now acceptable. I think that the arguments and derivations given by the authors can cause an interesting discussion among specialists. Therefore, the article can be recommended for publication.

Author Response

Dear Referee

We thank you for careful refereeing. And we are happy that the last submited version suits you.

Sincerely yours

Reviewer 2 Report

The paper presents an asymptotic view on the flow in shallow water, to include effects of viscosity and rotation into the usual inviscid flow models (Euler). In this revised version of the manuscript the presentation is much better understandable for (possibly) interested readers. For instance, the Introduction has been seriously enlarged to give a wider view on the why and how of the presented research. 

Line 86: Equation (2) surprises me. It has been changed from $\mu^-5$ to $\mu^-6$, whereas the text around it has been kept the same. Was the previous $\mu^-5$ a typo? Anyway, better explain where the $\mu^-6$ comes from? I did see that the definition of Re has changed, so this could make the difference.

Lines 92-99: I am glad to see that the highly confusing discussion of the other Reynolds number R has disappeared, making this part of the text better digestible.

Page 4: Figure 2 now gives a clear picture of the two-layered structure and of the coordinate systems in vertical direction (e.g. the range of \gamma and the role of "-1"), which helps to understand the matching activities further on in the manuscript. In the previous version of the manuscript this matching was quite 'magical' for me. 

Line 107-109: The delicate aspect of Lemma 1 is now better explained.

Page 9: I also noticed some more explanatory new text at the end of Section 2.

Page 10: Some more useful text has been added in the section on Conclusions.

Line 182: When I asked for a possible link with Stewartson's triple-deck theory, I had the impression from the text that the authors were studying a three-layered asymptotic structure.  Stewartson's triple-deck is the most famous of those structures. Now that the authors have better explained the asymptotics (around Fig. 2) it is clear that their structure is only two-layered, and the triple-deck is not related at all. So it makes no sense anymore to look for a connection.

In summary: The text has definitely improved, and readers will now be able to follow the broad lines. With some minor editing (see above) the manuscript could be published.

Author Response

Dear Referee,

First I want to thank you for refereeing and spending time to check. Below, I let only the suggestions and (constructive) critics:

> Line 86: Equation (2) surprises me. It has been changed from $\mu^-5$ to $\mu^-6$, whereas the text around it has been kept the same. Was the previous $\mu^-5$ a typo? Anyway, better explain where the $\mu^-6$ comes from? I did see that the definition of Re has changed, so this could make the difference.

You are right, the definition has changed because a referee prefered this definition (with the longer scale $l$ instead of $h$). So we updated all the formulas and there should be no typo. We increased the justification for non-specialists readers.

>Lines 92-99: I am glad to see that the highly confusing discussion of the other Reynolds number R has disappeared, making this part of the text better digestible.

Thanks.

>Line 182: When I asked for a possible link with Stewartson's triple-deck theory, I had the impression from the text that the authors were studying a three-layered asymptotic structure.  Stewartson's triple-deck is the most famous of those structures. Now that the authors have better explained the asymptotics (around Fig. 2) it is clear that their structure is only two-layered, and the triple-deck is not related at all. So it makes no sense anymore to look for a connection.

We agree and took this part off.

In summary, we fully agree with the comments and took the suggestions into consideration.

We do hope you agree with our modifications.

Sincerely yours

This manuscript is a resubmission of an earlier submission. The following is a list of the peer review reports and author responses from that submission.

Round 1

Reviewer 1 Report

The paper presents an asymptotic view on the flow in shallow water, to include effects of viscosity and rotation into the usual inviscid flow models (Euler). The presentation asks quite a bit from an unsuspecting reader. 

There is a small (d) vertical and a large (l) horizontal length scale defining a small parameter \mu = d/l with a Reynolds number Re based on the small length scale. Usually the Reynolds number is based on the large scale, and here the confusion starts. 

So, the first problem arises in equation (2), where the scaling Re ~ \mu^{-5} is mentioned. The thickness of the viscous layer is given by (3) as \mu^2. A reader does want to know where these scalings come from. In the lines that follow a classical Reynolds number R is introduced; the thickness then becomes R^{-1/2} as usual. It is explained that the difference in Reynolds numbers comes because a different non-dimensionalization leads to an unusual scaling of the equations (1), if I understand the text correctly. This text (above 2.1) is quite confusing, and I wonder why the classical Reynolds number is not used throughout. So explain first why this confusing Reynolds number Re must be used. Switching to R would keep many readers on board.  

Is there a relation with Stewartson's triple deck (developed in the early 1970s). There the lower layer has thickness R^{-5/8} while Prandtl's equations govern the flow. On top of that is an inviscid middle layer of classical BL thickness, and outside is a potential-flow region where the flow is elliptic. The triple-deck describes local irregularities in a BL (separation, trailing edge, shock wave). Is there a link?

On page 3+4 an integration in vertical direction is employed, which at first sight seems standard, but a new symbol \gamma_\infty pops up. Where does it come from? What is its purpose? Would a picture of the layers help to point at the limit y = -1+\mu^2\gamma_\infty? Does y=-1 correspond with the bottom and \gamma_\infty with the edge of the boundary layer? Are there two or three layers above each other? Also the other limit of y=\epsilon\eta should be explained. Is it the 'bottom' of the outermost inviscid region?

On page 4 Lemma 1 is introduced, which seems quite obvious. Why is the interchange of differentiation and integration not trivial? Are there terms which are singular, so that the integrals may not exist? Why is the vanishing of F at y=\epsilon \eta important;  is it because the integration interval becomes infinitely large? The unnumbered equation on top of page 5 comes completely out of the blue, further enlarging the confusion. Without understanding what is going on, I can no longer follow the reasoning from (11) onwards.

Neither do I understand what seems to be the major worry of the authors. Why do they want to further simplify the equations? Prandtl's equations are not that difficult to solve, at least when there is no flow separation. So, the overall goal of the paper is also unclear to me. Please, rewrite the manuscript in a way that readers can understand.  

Reviewer 2 Report

In the paper under review, the authors discuss the derivation of so-called viscous Serre-Green- Naghdi system by using both the inviscid Euler equations and the Navier-Stokes equations. An important feature of the work is that the corresponding computations are performed in the fully nonlinear regime. The authors focus on an unsolved problem: "the nonlinear viscous case lags behind the linear viscous case and the derivation is currently blocked at the Prandtl’s step. Would it be possible to further simplify such a coupled Prandtl system similarly to the classical one?"
In general, the paper may be of interest, but in the present form seems a preliminary draft. In my opinion, the authors analyzed too few literary sources (only 7), focusing mainly on work [5]. It is necessary to look at this problem in a broader way and analyze a larger number of literary sources. This will also eliminate another drawback of the manuscript, which is the small volume of the manuscript.
Minor comments:
1) It is advisable to introduce a nomenclature to make it easier for the reader to follow the reasoning and formulas.
2) Need to clarify the term "a characteristic horizontal length" (see Lines 39-40).
3) It is necessary to explain how formula (2) arises (see Line 54).
4) It is desirable to include the slip boundary condition in the reasoning as a possible case